# Food Intake Actions Detection: An Improved Algorithm Toward Real-Time Analysis

**DOI:** 10.3390/jimaging6030012

**Published:** 2020-03-17

**Authors:** Ennio Gambi, Manola Ricciuti, Adelmo De Santis

**Affiliations:** Dipartimento di Ingegneria dell’Informazione, Università Politecnica delle Marche, 60131 Ancona, Italy; m.ricciuti@staff.univpm.it (M.R.); adelmo.desantis@staff.univpm.it (A.D.S.)

**Keywords:** food intake monitoring, depth frame, Kinect, real time

## Abstract

With the increase in life expectancy, one of the most important topic for scientific research, especially for the elderly, is good nutrition. In particular, with an advanced age and health issues because disorders such as Alzheimer and dementia, monitoring the subjects’ dietary habits to avoid excessive or poor nutrition is a critical role. Starting from an application aiming to monitor the food intake actions of people during a meal, already shown in a previously published paper, the present work describes some improvements that are able to make the application work in real time. The considered solution exploits the Kinect v1 device that can be installed on the ceiling, in a top-down view in an effort to preserve privacy of the subjects. The food intake actions are estimated from the analysis of depth frames. The innovations introduced in this document are related to the automatic identification of the initial and final frame for the detection of food intake actions, and to the strong revision of the procedure to identify food intake actions with respect to the original work, in order to optimize the performance of the algorithm. Evaluation of the computational effort and system performance compared to the previous version of the application has demonstrated a possible real-time applicability of the solution presented in this document.

## 1. Introduction

Nutrition is an essential health determinant and affects the entire populations aging process [1,2]. In developed countries, the percentage of elderly is constantly growing [3] due to the achievements in medicine, technology and improvements of quality of life [4]. Direct consequence of the elderly population increase will be a growth in costs in the health system, which will have to be addressed by governments in the coming years [5]. The trend is further strengthened by the low level of births, which will lead to an increase in the ratio of elderly and active population in 2060 from 17.4% to 29.5% in Europe, from 13.2% to 21.9% in the United States, from 22.7% to 35.1% in Japan and from 8.2% to 29.5% in China [6]. The aging population will lead to an increase of diseases such as neuronal disorders, including Alzheimer’s and Parkinson’s [7].

To tackle this demographic challenge, actions must be taken, which can also make use of technologies. In particular, different services and devices could be introduced at home, to increase the comfort and promote the older people’s independence [8]. This approach will allow the elderly to live at their homes without the need of hospitalization, with technological aids helping to overcome limitations and inabilities caused by aging. Furthermore, the technology is a way to support caregiver activity [9], through the possibility to monitor the person’s status.

One of the most significant applications of this paradigm is ambient assisted living (AAL). The main activities of an AAL system [10] can be described as follows:*Health and protection*: to monitor the elderly by actions and activities recognition during all the day [11];*Privacy and security*: the system should be as “invisible” as possible, respecting the people’s privacy [12];*Communication and social environment*: the possibility of interaction with other people should be guaranteed both in indoor and in the outdoor living environments [13];*Mobility support*: an AAL application should also support the elderly outside their homes by helping them orienting in the city streets [14].

Among available devices on the market, two main categories stand out: contactless and wearable devices. Wearable techniques consist of the application of some specific sensor above the clothes or in contact with a subject’s skin to analyze and extract specific parameters and features. A common problem about the use of wearable technologies is given by the subject not accepting the device or forgetting to wear it. Nowadays, it is a priority to develop solutions that are increasingly compact and less invasive and/or unobtrusive [15].

The contactless technique relies on the fixed installation of sensors [16] inside the living environments of the subjects that adopt them. In this work, the Kinect, which embeds many sensors, such as an RGB camera, depth sensor and real-time skeleton acquisition [17], was used, thanks to its accuracy, low-cost and high level of availability [18].

By acquiring both visual information (RGB) as well as depth information (D), the RGBD camera allows us to extract a 3D representation of the objects captured in the scene. Privacy is not a concern as face details may be deleted without losing useful information for user’s motion or activity recognition [19]. Moreover, the information provided by the depth frame is helpful to overcome the light dependency that is typical of RGB cameras.

Nevertheless, contactless systems may not be the best choice in an outdoor environment [20]. A solution could be the data-fusion approach [21], in which contactless and wearable devices are used simultaneously. Combining the information provided by wearable and contactless devices, it is possible to improve the desired application performance.

In order to create an adaptive model approximating the human joints of the torso and arms, useful to track the food intake actions, in previous works [12,22], a neural network with unsupervised learning was implemented, based on the self-organization principle. The self-organizing maps extended (SOM Ex) [23,24] method was considered, which relies on a neural network to model the human brain performing a specific operation [25,26].

In the present paper, three novelties are introduced. The first is the automatic approach to set the start frame (SF) and the end frame (EF), which correspond to the beginning and to the end of the meal, the second is an improvement of the SOM Ex prediction to avoid the union of the adaptive model joints and the third is the management of data storage using a new software interface.

By introducing these three processing phases, the computational time of the system decreases, thus representing the basis of an effective real-time implementation.

In a recent study [27], the authors proposed a deep bilinear learning method as a high performance actions recognition system. They exploited RGB, depth and skeleton sequences demonstrating efficacy of learning time-varying information in their results. With respect to the present paper, the depth frames are investigated to track the subjects movement using an adaptive model in order to preserve their privacy in case of domestic installation of the system. RGB stream is only used as a ground-truth, allowing us to compare the resulting actions number during a meal.

In another study [28], the authors employed YCbCr and YIQ color spaces as the transformation of RGB, to monitor subjects with Alzheimer or dementia during the food intake actions from a frontal view. They then analyzed skin regions to detect the actions with promising results, but did not permit the preservation of data due to the privacy of the subject.

In 2014, Tham et al. [29] investigated an automatic detection method to extract drinking activities by using depth data from RGB-D camera, like the the sensor used in the presented work. As they stated, by using depth data, the problems related to change of lighting conditions can be prevented providing a greater system efficiency and accuracy.

In the same year, a study described in [30] analyzed the movements of the elderly eating and drinking. A young man, performing the same activities, is used as a reference. The authors, in order to estimate the gestures, focused on the head and hand joints, defining the distance from each hand to the head as the discriminating parameter. This approach differs from the study proposed before, as the Kinect was placed in front of the subjects and the RGB stream is employed to extract their skeleton.

If the privacy of subjects is not a priority, exploiting the RGB information could be useful to monitor and discriminate the subjects’ hand and face gesture while eating, like in [31].

In a work presented in 2016 [32], the authors used a portable wearable device, implemented in eyeglasses, linked to a piezoelectric strain sensor for food intake and activity recognition. Even though the authors distinguished the food intake actions between other not eating actions, with a average F1-score of 99.85%, subjects feel less comfortable when they have to remember to wear a device.

In a recent paper [33] RGB and depth image pairs acquired before and after a meal are processed in order to monitor hospitalized patients by estimating nutrient intake volume, calculating the corresponding weight and then converting to the calories by using a food calorie database. They employed a novel multi-task neural network to allocate a great number of meals for training: 30 meals for validation and 60 meals for testing. They calculated the system performance by the mean absolute error (MAE) and mean relative error (MRE) as indices of the system precision [34,35].

An interesting study [36] that could represent a possible additional analysis for the present paper, processed the depth frame to recognize the contour of dishes on a dining table by carrying out a normalization. The depth frames are implemented as packages of robot operating system (ROS), in order to work on a mobile robot useful to pass medicines to the patients.

This paper is organized as follows: Section 2 describes the algorithm based on the SOM extended implementation developed for the top view sensor configuration. Section 3 describes the main steps for data management and Section 4 shows the optimization introduced in the present work. Section 5 presents the novel approach by describing the main algorithm for the identification of the start and stop frames and Section 6 shows and discusses the results. Finally, in Section 7, the main conclusion of the work is reported.

## 2. The Original Algorithm

The food intake detection algorithm considered here relies on an unobtrusive method, in which a Kinect v1 device is placed on the ceiling of the two closed rooms with the aim to reproduce the same set-up in a domestic environment, avoiding the acquisition of the subjects’ faces and preserving their privacy. In order to evaluate the algorithm performance, many tests were performed by seven subjects (two females and five males) with varying durations, table positions and the person’s upper part of the body orientation with respect to the table. The subjects taking part to the tests were between 22 and 52 years old, with heights in the range of 1.62 to 1.97 m. The experimental set-up is shown in Figure 1.

The depth information, acquired with a frame rate of 30 fps, let us calculate anthropometric features and the distances between the head of the subject and the Kinect device in order to create the input skeleton model for the SOM Ex algorithm. The Sobel edge detection is exploited to find the borders of the objects in the depth image and to distinguish the person from the table.

The 3D map representation of the person’s blob is called the point cloud (PC). The preliminary processing [12] to extract the PC consists of the following steps:The depth frame acquired by the sensor is inserted into a matrix, where the sensor-object distance is associated to each pixel.The “holes”, i.e., the zeros of the matrix containing the depth frames, within the image are appropriately filled.A reference depth frame, which consists of the background only, is subtracted from the current depth frame in which the person is in the scene. The subtraction result is the PC.Inevitably, the device introduces noise, especially in those points where there are large depth jumps within a few pixels of distance, such as on the table edges or on the border of the field of view. In order to remove the noise that affects the depth frames, a median filter with a 5 × 5 pixel window is used.Finally, the depth camera parameters can be extracted and the person’s PC calculated.

As previously introduced, the input model for the SOM Ex algorithm is made up of 50 points that provide the position of the joints and are the neural network nodes.

With a proper choice of the depth camera parameters [37], the depth blob is transformed in a PC (Figure 2) that is used to fit the input model to the SOM Ex.

The algorithm allows us to generate the head nodes positions at the highest points of the PC, separated by the torso nodes, which are rotated according to the geometrical orientation of the person with respect to the table. The three most important nodes are the joints $J_{\mathrm{hd}}$, $J_{\mathrm{hl}}$ and $J_{\mathrm{hr}}$, marked in the Figure 2, corresponding to the head center and to the hands. These nodes are checked to avoid an undesired union between them. Furthermore, thanks to the adaptive model, it is possible to track these joints to identify the SF and EF, in which food intake actions happen and to calculate the number of actions performed by the subject. The goal is to process the person’s PC to find the head joint $J_{\mathrm{hd}}$, left hand joint $J_{\mathrm{hl}}$ and right hand joint $J_{\mathrm{hr}}$ (see Figure 2). In particular, the SOM Ex traces the movements of the subject using a model with 50 nodes, for a total of 47 plans, 9 of which are used to model the subject’s head, as shown in the Figure 3. The number of nodes is determined empirically to reach a good accuracy in person’s tracking [22]. Like Figure 3 shows, the input PC fits the predefined skeleton model, providing in output the adapted model for each processed frame.

### Limits of the Food-Intake Recognition Algorithm

The SOM Ex algorithm developed in [22] has proved to be efficient in the subject’s movements tracking. However, the tests have been conducted on a limited number of frames (500–1000) not corresponding to a real case which is much longer in time and requires the processing of a larger number of frames.

Moreover, to process such a great number of depth frames, a control mechanism that avoids the union of the $J_{\mathrm{hd}}$, $J_{\mathrm{hl}}$ and $J_{\mathrm{hr}}$ is mandatory. In fact, the past implementation of the algorithm using SOM Ex has shown tracking errors due to network nodes attracting each other during the processing, making it necessary to introduce a stabilization strategy in order to improve the SOM Ex algorithm prediction.

## 3. Data Management

The application software for data management consists of three steps:Kinect software development to store data;tracking control;post-processing functions.

The first step is related to the development of a software that is able to speed up the processes of continuous depth and RGB data acquisition and storage. Figure 4a,b show the software interface in cases of “background” and “person in the scene” acquisitions respectively.

The developed software allows us to buffer the background data (the scene without the subject). When a subject enters the field of view of the Kinect, an ad-hoc function detects the persons presence and the software interface shows a green square on the subject’ head (Figure 4b). With the person in the scene, the background with foreground (B+F) data are buffered. The data previously stored in the buffer (background image) are saved on the disk. When the subject leaves the scene, the buffer is cleared and the software automatically stores depth and RGB data on the computer disk. If there is no one present in the scene, the buffer is cleared every 10 min and then filled again.

Once data are saved, depth frames are processed. The tracking control adapts the model to the movements of the person, implementing the SOM Ex algorithm.

Taking into consideration the anthropometric measures of the subject under test that constitute the adaptive model, the post-processing functions trigger the calculation of the Euclidean distances between head and hands joints and checks that they remain separated. The control consists of introduction of thresholds and a reset mechanism. In particular, if the $J_{\mathrm{hd}}$ and $J_{\mathrm{hl}}$ (or $J_{\mathrm{hr}}$) shows a Euclidean distance which is less than 300 pixels, for more than five seconds (threshold-time corresponding to 150 frame, because the processing frame rate of 30 fps), a reset mechanism restores the initial model which fits the subject’ current movement instantly. The 300 pixel value corresponds to about 200 mm and was empirically determined by measuring hand and mouth distance from the ground-truth, which is represented by the RGB frames. We set experimentally these thresholds considering the situations consistent with a normal drinking phase, which is longer than a food intake action.

## 4. Optimization of the Algorithm Implementation

To be in line with a usual meal, the goal of the food-intake detection proposed in the present paper is to reach 15 min of acquisition (equal to 27,000 frames at 30 fps). Furthermore, we considered that the duration of the experimented dietary habits matching with a normal meal. After every 15 min sequence, the developed system re-initialize and then drops to a new file with a new file name.

The action of the food-intake is associated with the distance between the joint of one or both hands with respect to the head, which must be less than a predetermined threshold value, provided that this distance is maintained for at least 5 s. In this way we avoid confusing the action of food intake with other gestures for which the hands approach the mouth for limited time intervals.

However, during the processing of a large number of frames (tens of thousands), the previous algorithms [12,22], before carrying out the appropriate correction, shows unwanted situations, such as the loss of the joints of the hands that remains blocked at the joint of the head. With the new implementation of the algorithm, the loss of joints forming the anthropometric adaptive model is corrected, by restoring the initial model for SOM Ex each time a fusion between joints is recognized. This allows the SOM Ex to start again and instantly stabilize to track the next movement. The continuous restarting of SOM Ex makes the algorithm more accurate than the previous implementation. We also considered the total time spent to process the entire actions detection.

## 5. Identification of the Start and End Frames

The algorithm block diagram to automatically set the SF and the end frame (EF) is shown in Figure 5.

The SOM Ex algorithm is used to trigger data storage by recognizing the moment that the subject is seated with one hand on the table. The functionality to extract the start frame (SF) is introduced in order to identify the initial frame that a person sits at a table, using a low-complexity and instantaneous person detection algorithm. The approach is based on a series of events checks; if one of these does not occur, it is not possible to proceed with the next step. The checklist is:extracting the PC by the person’s blob from the depth frame, like Figure 6 shows;check threshold to identify when the the subject is beside the table with a definite distance, like shown in Figure 7;the threshold on a parameter, called “Ratio”, identifies when the person is sitting at the table, visible in Figure 8. The Ratio parameter will be described in detail later;the recognition of the person sitting with one hand on the table occurs when the control thresholds “hand gap” and “hand blob” are met, like it shown in the Figure 9.

The acquisition software developed is able to acquire background frames at defined intervals when the person is not in the scene. This issue is very important because it offers an updated background. In fact, the person is free to decide when and how to sit, also by moving the chair to any side of the table. Subtracting the background (*B*) frames from the depth data (B+F), allows us to highlight the foreground (*F*). In this way, the person’s blob is extracted and transformed in PC through the intrinsic parameter of the depth camera. In order to evaluate if the person has approached the table sufficiently, the maximum midpoint on the person’s head inside the table area is considered.

By defining “Dist” (= 45 pixel) as the range of the coordinates (x, y) of a box around the table (Figure 7), the first control is the position of the person to ensure that the subject is beside the table. If the distance between the person and the box is greater than “Dist”, the algorithm does not proceed to the next control.

To identify the person sitting, the distance between the sensor and the head of the person is considered, taking into account of the distance values in the previous frames. An empirical parameter Ratio is defined. The Figure 10 shows the meaning of the Ratio parameter. Let the sensor be positioned at 3000 mm above the floor level at the instant t=0 s the distance of the person’s head is p=1100 mm from the sensor, and we can estimate the subject height by calculating the difference between sensor height (3000 mm) and head height (1900 mm), which results in 1100 mm. The table is located 2100 mm from the Kinect sensor. The chosen value for the Ratio parameter is important because every single person may be taller or shorter than another, so interpreting if the person is sitting could be difficult and creating ambiguity. To resolve ambiguity, we store the head blob maximum height value, as measured in each frame, in a matrix. In this way, it is possible to log a history of events that occurred in the time-frame before the current one. During the tests performed by each subject of different height, we found empirically that a person standing reduces the distance from the sensor height by about 30% with respect the sitting position.

Based on this consideration, the threshold value Ratio is set to 0.3. The position is then classified as “sitting” when the distance from the Kinect sensor increases more than the Ratio percentage value with respect to the distance that was detected in the “standing” position.

When the person is seated, it is necessary to analyze the hands’ correct position, which must be on the table area, to represent an input model for the SOM Ex. The table area, saved in a matrix as a binary image, is multiplied with the matrix that represents the current foreground frame composed of different blobs (head, hands, ...). To isolate the hands blob from the others, the algorithm extracts only the points above to a threshold height, defined as “hand gap” from the table. The introduction of the “hand gap” threshold is necessary because when a subject is close to the table and is about to sit, he/she tends to protrude the trunk forward, which for a few moments is over the table. When the subject completes the movement and sits on the chair, the algorithm starts to process the correct input model. Considering that the hands must be closer than 200 mm to table to trigger the event, the “hand gap” threshold is set at 200 mm in height. Finally, the hands blob are identified with a hand area size threshold set to 300 pixel to indicate that a blob greater than 300 pixel is probably a hand on the table and the person is actually sitting with his hands, or at least one, at the table.

Once the algorithm carried out all the checks against the parameters set as thresholds, an ad-hoc developed function finds and stores the SF and the EF (i.e., the frame that the subject rises from the table or the last frame of the file), in a dynamic matrix called “SetParam”. The dynamic matrix is built, adding a column every time a new test is processed and the result is a matrix with as many columns as the number of tests and three rows. The first row is the test number (1st row), the second is the SF correspondent (2nd row) and the third is the correspondent EF (3rd row) (Table 1).

### Management of Multiple Start and End Frames

When the person gets up and moves away from the table for a short time, we consider the same meal to continue, even if there are multiple SF and EF (see Table 2). In this case it is necessary to identify the frame in which the person is sitting at the table, like that shown in Figure 3, gets up and then returns to sit down, taking the suitable position to restart the SOM Ex algorithm. To obtain an efficient system, the algorithm should avoid processing frames in which the subject is no longer in the scene because he has risen from the table. The approach used is to check different conditions and set thresholds to detect when the person ends the meal and rises from the table.

## 6. Results and Discussion

As previously stated, the purpose of the present work is to introduce suitable corrections to the previously proposed algorithm for the recognition of food intake action, in order to remove its limits and to constitute the basis for a real time implementation. In fact, the implementation described in [12,22] required a manual setting of the SetParam matrix. The SF and EF selection was manually carried out by an operator who, by examining all the frames, identified the first one in which the subject appears sitting with the hands on the table and the last one in which the subject finishes the test procedure.

Table 3 shows the comparison between the values of the SetParam matrices extracted in the two different ways, according to both an automatic approach and manual approach with the previous method [12,22]. In order to evaluate the performance of the two different solutions, the RGB videos sequences are used to compare the ground-truth (GT) values of the actions number estimated by applying the SOM Ex algorithm. Table 3 shows the 33 tests performed by seven subjects.

In Table 3, assuming that the manual operator was able to correctly identify SF and EF, the automatic algorithm achieves almost the same results with respect the GT. The mean absolute error (MAE) and mean relative error (MRE) are calculated in order to evaluate the system performance. The corresponding errors are reported in Table 4. In order to calculate the absolute error (AE) the following equation was considered
(1)$$AE=|\hat{x_{t}}-x_{t}|$$
where $x_{t}$ is the true value and $\hat{x_{t}}$ is the measurement. The MAE is the average of all absolute errors is calculated as follows:(2)$$MAE=\frac{1}{n}\sum_{t=1}^{n}\left|\hat{x_{t}}-x_{t}\right|$$
and the percent MRE, the average of all relative errors, is the following
(3)$$MRE\%=\frac{1}{n}\sum_{t=1}^{n}\left|\frac{\hat{x_{t}}-x_{t}}{x_{t}}\right|\cdot 100$$

In Table 4 it can be observed that MRE, calculated for the start frame manually set (SFM) approach, is the worst result. The automatic approach improved considerably the algorithm performance even though the MAE is slightly higher in the automatic approach for the SF detection.

The system errors with respect to the ground-truth especially occur in the following tests:In tests 4, 13 and 21 the height of the person was not correctly detected because the person entered the scene and sat down at the table too fast with respect to the time that the sensor needed to recognize the head of the person: the identification of the sitting action did not happen correctly.Test number 11 could not detect that the subject sat down at the table since its position was on the border of the frame (see Figure 11) which meant that a correct SF could not be obtained. However, the algorithm provided SF = 1, corresponding to the beginning of the acquisition of the sequence, and therefore no data were lost regarding the intake actions;Test 18 showed an error in the EF setting, due to a particularly noisy acquisition, despite the median filter application, since the person was moving away from the table. The number of lost frames was insignificant in regards to counting the number of intake actions.

In order to provide evidence for the precision of the automatic SF and EF extraction while varying the value of the Ratio parameter, Table 5 shows the values obtained with Ratio equal to 0.2 with respect the ground-truth.

In Table 5, the comparison with the results of Table 3 shows all the improvements in tests 4, 13 and 21; they are almost perfectly aligned with manual values. On the other hand, there is an important performance decrease in sequence 22, which presents a large number of frames; if Ratio is set to 0.2, the algorithm becomes more sensitive to the subject’s height detected by the Kinect device. Moreover, in Table 5 two errors remain in test 18, which is related to the quality of the RGBD stream acquisition and can hardly be corrected; the algorithm is however able to find a SF without loss of information. It is clear that there is a trade off between the two choices of the parameter Ratio: on the one hand, with a higher value of Ratio, sequences with a large number of frames are privileged. On the other hand, seat errors are corrected by leaving lower values of Ratio, but at the same time, the algorithm is more sensitive to fluctuations in the height of the subject, with the consequent detection of false positives. Not considering the sequence 22 and calculating the MAE and MRE, the system performance in Table 5 improves with respect to Table 3 as regards both the MAE and the MRE for SF detection results. The corresponding errors are reported in Table 6.

As a conclusion, the choice of the parameter Ratio must be made considering the properties of the acquired data. Since we are interested in monitoring the intake action for about 15 min, we used a more conservative value Ratio = 0.3, making the sequences of 20,000 frames more stable, is preferable. The actions detected with respect the Ratio = 0.3 are given in Table 7.

The errors of the actions detected in Table 7 with respect the ground-truth RGB video shown in Figure 12, resulted in a **MAE** of 0.424% and a **MRE** of 5.044%.

## 7. Conclusions

The present work represents an extension and optimization of a previous study already published in the literature, whose purpose was the automatic depth data storage and the automatic recognition for the food intake actions monitoring performed by persons sitting at the table. This novel approach increases the system efficiency through the introduction of an algorithm able to recognize the start and end frames of the useful sequence by exploiting the RGBD stream. At the same time, the automatic data storage provides an optimization of the food intake action recognition algorithm, since it allows us to solve the problem related to the processing of very long sequences, also thanks to being able to avoid saving entire sequences. With the new implementation, a 15 min sequence, corresponding to 27,000 frames, was processed in about 30 min, using a processor i7 4470 K with frequency 3.5 GHz and RAM 16 GB. Prior to the algorithm optimization, the required processing time was of several hours for the same number of frames processed. The results obtained show a great effectiveness of the automatic detection algorithm of SF and EF, and a considerable reduction in processing times. In this way the foundations were laid for the realization of a completely automatic application capable of processing the results in real time.

## Figures and Tables

**Figure 1 jimaging-06-00012-f001:**
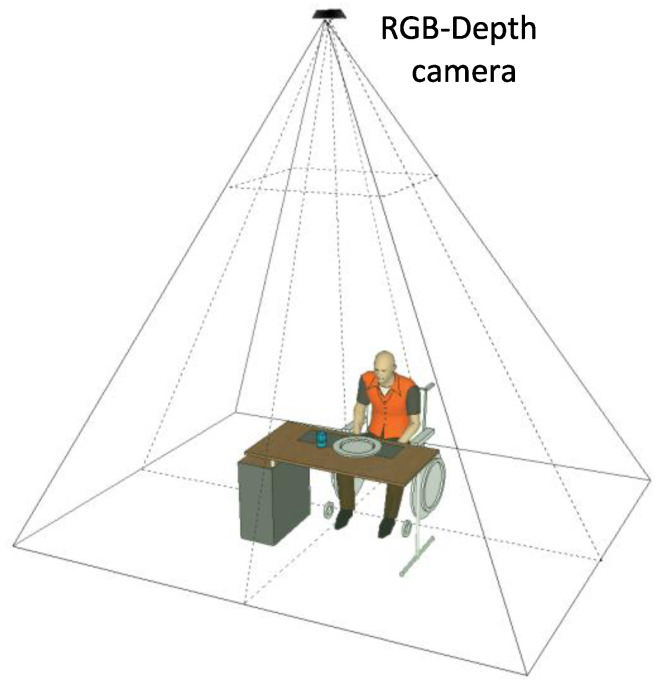
The system top view configuration.

**Figure 2 jimaging-06-00012-f002:**
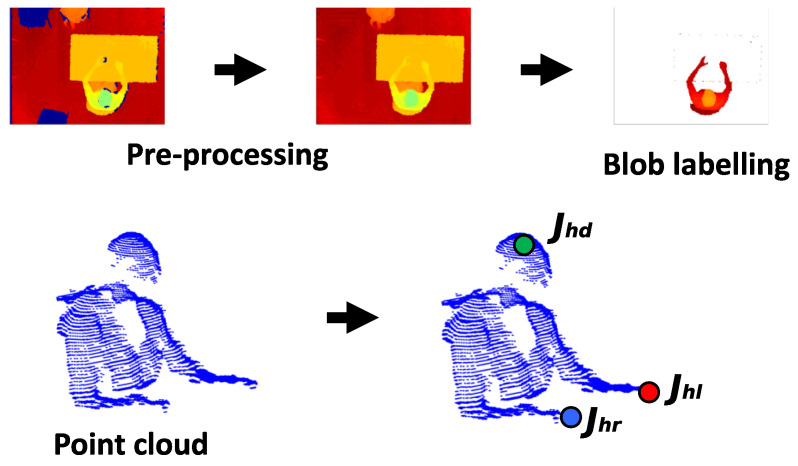
The point cloud (PC) extraction process and identification of the input model principal joints: $J_{\mathrm{hd}}$ is the center of the head joint, $J_{\mathrm{hr}}$ is the right hand joint and $J_{\mathrm{hl}}$ is the left hand joint.

**Figure 3 jimaging-06-00012-f003:**
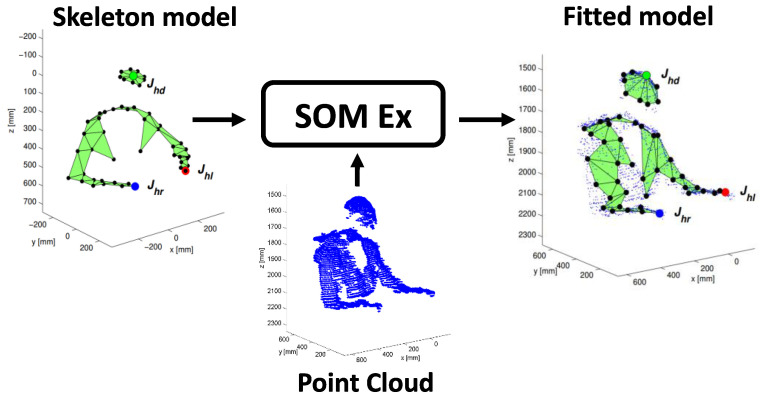
The self-organizing maps extended (SOM Ex) algorithm produced the fitted model as the tracking result, starting with the initial model and the PC.

**Figure 4 jimaging-06-00012-f004:**
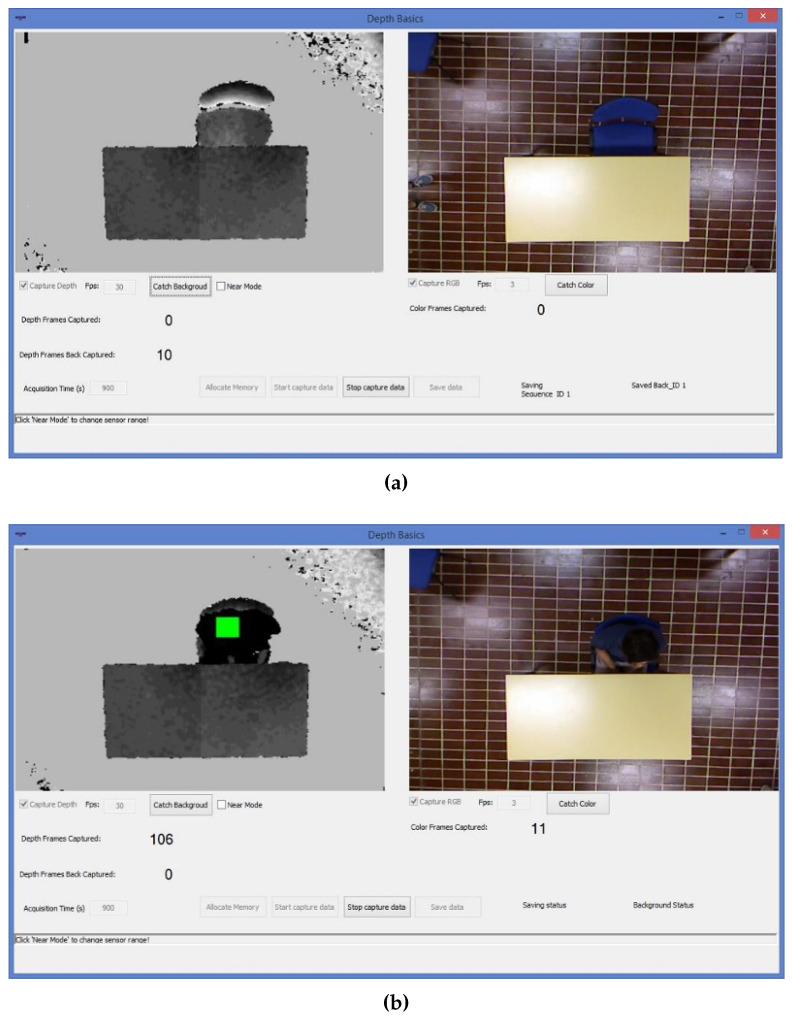
(**a**) Software interface developed. On the left side, the depth frame is visible, On the right side, the RGB color frame. The background is captured and stored in a buffer. (**b**) Depth and RGB data capture during the test. The buffer is filled with the background data, but is emptied when the subject is present in the scene and the background data is stored on disk simultaneously. Then the buffer fills up of background and foreground (B+F) data as long as the subject is in the scene. The data of B+F are memorized when the subject leaves the scene, at the same time, the buffer of B+F is emptied.

**Figure 5 jimaging-06-00012-f005:**
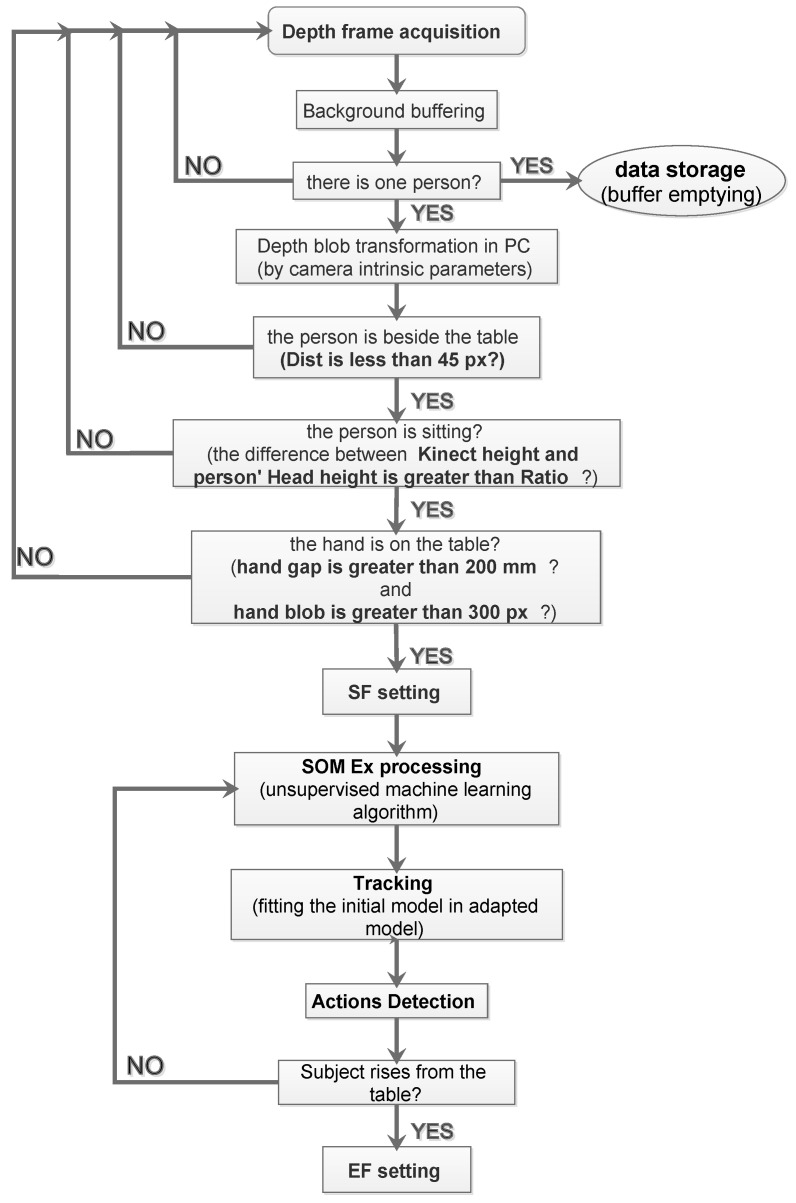
Algorithm block diagram to extract start frame (SF) and end frame (EF).

**Figure 6 jimaging-06-00012-f006:**
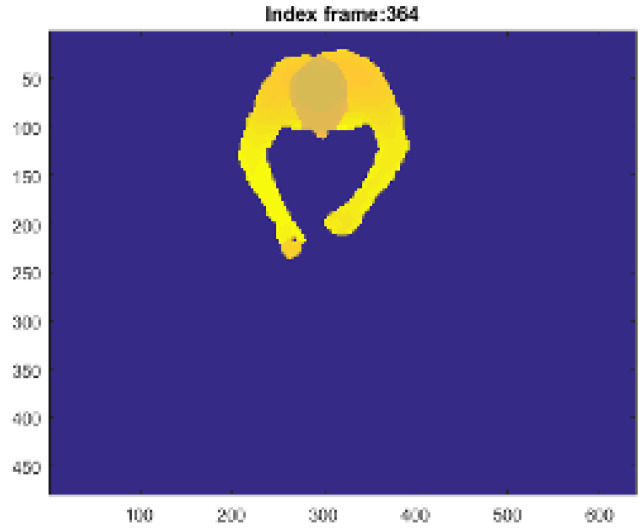
The PC is extracted subtracting the person’s blob from the background depth frame.

**Figure 7 jimaging-06-00012-f007:**
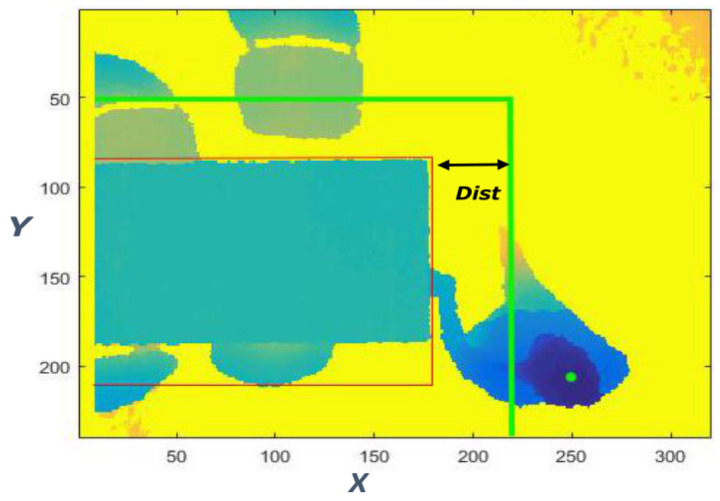
The person’s distance from the table is considered as an event to set the SF.

**Figure 8 jimaging-06-00012-f008:**
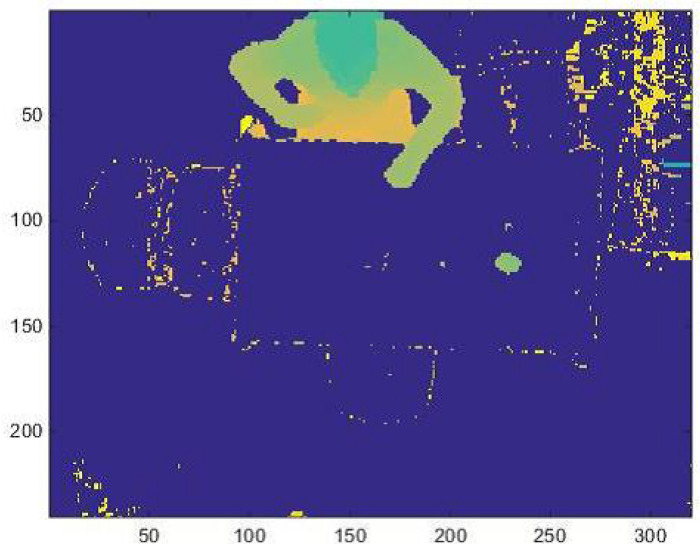
The algorithm extracts the person sitting at the table.

**Figure 9 jimaging-06-00012-f009:**
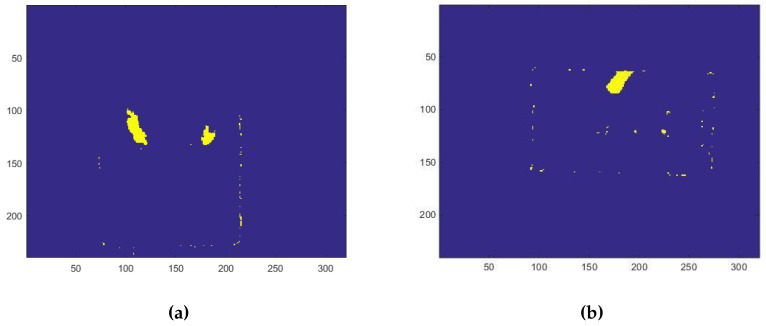
(**a**) The blob of both hands are extracted; (**b**) only one hand’s blob of the person sitting is extracted.

**Figure 10 jimaging-06-00012-f010:**
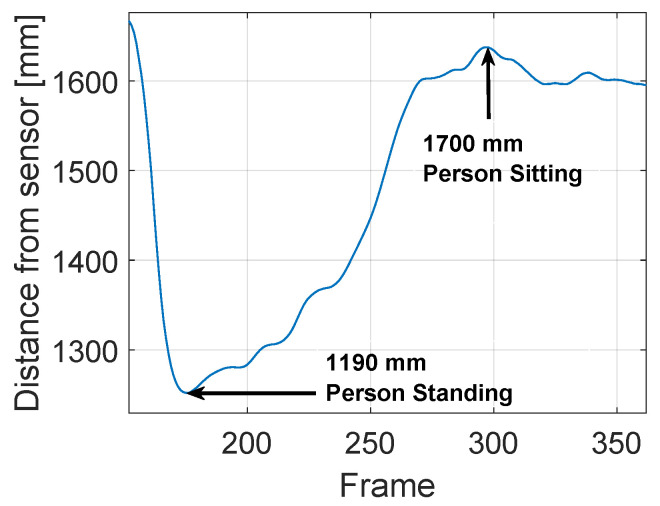
The point in the minimum peak of 1190 mm indicates the person standing identification with minimum person’s distance from the sensor. The frame in which the person change his/her position from standing to sitting, with a consequent increase in the person’s distance from the sensor, is indicated in the maximum peak of 1700 mm. The Ratio is calculated as 30% less than 1700 mm, that is 1190 mm.

**Figure 11 jimaging-06-00012-f011:**
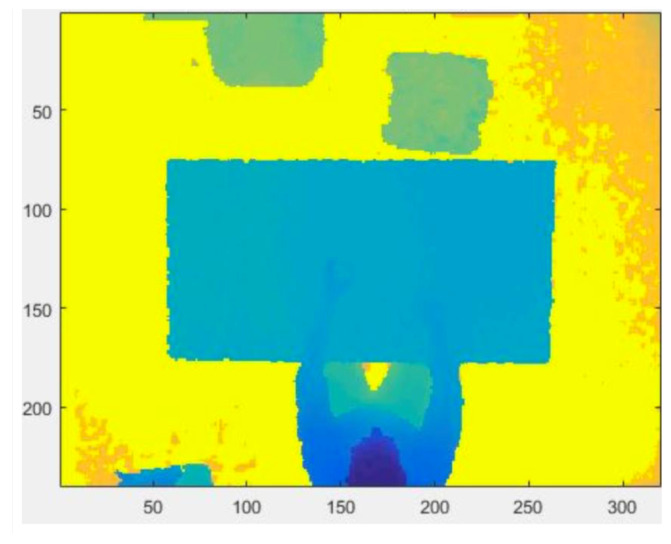
A position of the subject not completely inside the frame cannot allow the detection of SF.

**Figure 12 jimaging-06-00012-f012:**
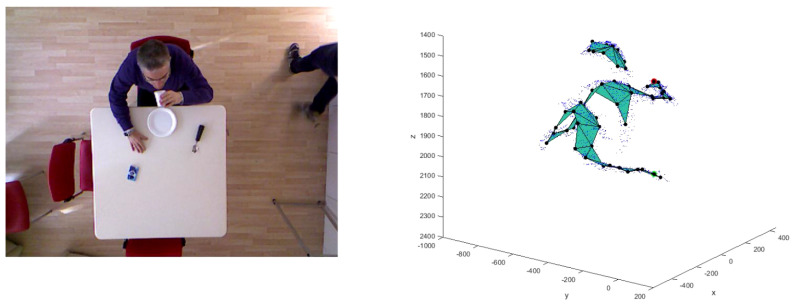
The ground-truth for the actions detected is represented by the video RGB matching.

**Table 1 jimaging-06-00012-t001:** Example of “SetParam”, 3×n matrix, where *n* is the test number processed. Only 7 columns of processed tests are visible. The rows respectively indicate the test number (TN), the start frame (SF) and the end frame (EF). The columns represents the sequential number of tests processed.

Set Param (3xn)							
**TN**	1	2	3	4	5	6	7
**SF**	215	170	200	179	250	204	148
**EF**	500	500	500	500	500	500	500

**Table 2 jimaging-06-00012-t002:** SetParam matrix with multiple SF and EF; two sequences of acquisitions during the tests are highlighted: the 11th and 13th are divided into two pairs of SF and EF. The 12th sequence, however, is constituted by a single pair of SF and EF.

**TN**	10	11	11	12	13	13
**SF**	0	277	7376	403	165	6019
**EF**	0	7024	20023	5390	5330	8052

**Table 3 jimaging-06-00012-t003:** Comparison between the automatic and manual SF and EF calculated with respect the ground-truth values identified using Ratio=0.3.

Subject	N. Test	Automatic Approach		Manual Approach		Ground-Truth	
		SF	EF	SF	EF	SF	EF
1	1	171	500	176	500	171	500
	2	149	500	152	500	149	500
	3	220	500	218	500	220	500
	4	211	500	179	500	186	500
	5	225	500	231	500	225	500
2	6	171	500	183	500	171	500
	7	100	500	113	500	100	500
	8	1	500	1	500	1	500
	9	84	500	88	500	84	500
	10	130	500	130	500	130	500
3	11	1	500	219	500	180	500
	12	93	500	110	500	93	500
	13	147	500	128	500	138	500
	14	145	500	151	500	145	500
	15	153	500	145	500	153	500
4	16	147	500	142	500	147	500
	17	207	500	208	500	207	500
	18	165	468	165	500	165	472
	19	113	500	113	500	113	500
	20	153	500	140	500	153	500
5	21	80	7011	221	7030	218	7011
	22	7376	20,023	7385	20,020	7376	20,023
	23	403	5390	403	5390	403	5390
	24	165	5330	161	5330	165	5390
	25	6019	8052	6019	8050	6019	8052
6	26	56	3599	90	3600	56	3599
	27	1	3899	10	3899	1	3899
	28	270	2639	300	2640	270	2639
	29	295	5400	300	5400	295	5400
	30	405	5381	417	5400	405	5381
7	31	158	5224	172	5286	158	5224
	32	6015	8048	6032	8100	6015	8048
	33	1	3599	8	3599	1	3599

**Table 4 jimaging-06-00012-t004:** System performance calculated with MRE using the Ratio=0.3; MAE—mean absolute error; MRE—mean relative error; SFA and EFA—start frame and the end frame, respectively, estimated with the automatic approach; SFM and EFM—start frame and end frame set with the manual approach.

	MAE	MRE
**SFA**	10.636	5.537%
**EFA**	0.121	0.026%
**SFM**	8.939	54.410%
**EFM**	5.667	0.257%

**Table 5 jimaging-06-00012-t005:** Comparison between the automatic and manual SF and EF identification calculated with respect the ground-truth values identified using Ratio=0.2. ND—not detected.

Subject	N. Test	Automatic Approach		Manual Approach		Ground-Truth	
		SF	EF	SF	EF	SF	EF
1	1	171	500	176	500	171	500
	2	148	500	152	500	149	500
	3	220	500	218	500	220	500
	4	176	500	179	500	186	500
	5	225	500	231	500	225	500
2	6	171	500	183	500	171	500
	7	98	500	113	500	100	500
	8	1	500	1	500	1	500
	9	82	500	88	500	84	500
	10	130	500	130	500	130	500
3	11	220	500	219	500	180	500
	12	91	500	110	500	93	500
	13	128	500	128	500	138	500
	14	153	500	151	500	145	500
	15	140	500	145	500	153	500
4	16	147	500	142	500	147	500
	17	197	500	208	500	207	500
	18	165	468	165	500	165	472
	19	113	500	113	500	113	500
	20	134	500	140	500	153	500
5	21	165	7011	221	7030	218	7011
	22	ND	ND	7385	20,020	7376	20,023
	23	403	5478	403	5390	403	5390
	24	165	774	161	5330	165	5390
	25	775	5763	6019	8050	6019	8052
6	26	56	3599	90	3600	56	3599
	27	1	3899	10	3899	1	3899
	28	270	2639	300	2640	270	2639
	29	295	5400	300	5400	295	5400
	30	405	5381	417	5400	405	5381
7	31	158	5224	172	5286	158	5224
	32	6015	8048	6032	8100	6015	8048
	33	1	3599	8	3599	1	3599

**Table 6 jimaging-06-00012-t006:** System performance calculated with MRE using Ratio=0.2; MAE—mean absolute error; MRE—mean relative error; SFA and EFA—start frame and end frame, respectively, estimated with the automatic approach; SFM and EFM—start frame and end frame, respectively, set with the manual approach.

	MAE	MRE
**SFA**	4.313	4.440%
**EFA**	2.938	0.078%
**SFM**	7.938	57.253%
**EFM**	5.750	0.265%

**Table 7 jimaging-06-00012-t007:** Number of actions detected (NAD) with respect the actions identified with the video RGB matching ground-truth (AGT) results.

Subject	N. Test	NAD	AGT
1	1	5	5
	2	7	7
	3	4	5
	4	3	5
	5	6	6
2	6	6	6
	7	3	5
	8	5	5
	9	6	6
	10	4	4
3	11	4	4
	12	5	5
	13	3	5
	14	7	7
	15	5	5
4	16	5	5
	17	5	5
	18	4	4
	19	7	7
	20	7	7
5	21	15	15
	22	27	32
	23	12	12
	24	15	15
	25	11	11
6	26	18	19
	27	19	19
	28	13	13
	29	16	16
	30	17	18
7	31	18	18
	32	11	11
	33	14	14

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
