# Peer review of "Food Intake Actions Detection: An Improved Algorithm Toward Real-Time Analysis"

_2313-433X, 2020, doi:10.3390/jimaging6030012_

Round 1
Reviewer 1 Report
Please revise equations (1), (2), (3). Define e_t? Define the 2 "x" symbols on Test 22, in Table 5. Please check the last sentence in Section 5. It is strange to review a manuscript “submitted to Journal Not Specified”.Author Response
Thank you for your observations.
We revised the three equations, changed the x symbol with ND (not detected), we also changed the last sentence in Section 5. We added the Journal of Imaging for the identification of the journal submition of the manuscript. We also checked extensively the manuscript and some expression has been corrected and highlighted with the blue color. We changed the Figure 10 because of the low resolution.
Reviewer 2 Report
The paper is somewhat incremental: it is an optimization of a previous approach, adding auto detection of Start and End frames of an action and management of data to deal with long data sequences.
The experimental setup is a bit dated: Kinect v1 used. This is not, in itself, a problem but some of the results may now not reflect precision and performance on hardware of today. Furthermore the state-of-the-art in off-the-shelf software solutions that comes with the new devices is also a lit better evolved. It was not clear to me why an overhead camera position is essential. From an angled view, current off-the-shelf solutions might be able to extract a reliable skeleton making pose metrics easier particular for sitting and standing detection.
The literature review seems adeqate as far as I can tell.
Various grammatical errors are littered throughout the paper.
Figure 10 is rasterized and a bit low quality.
Otherwise the paper presents some novelty, albeit in a very specific problem, and should be acceptable for publication.
Author Response
Thank you for your observations.
We are aware that Kinect v1 is an old device and we considered to use other devices of recent technology with respect Kinect v1 for future projects.
In order to clarify the concepts, we revised the three equations, changed the x symbol with ND (not detected) in the Table 5. We added the Journal of Imaging for the identification of the journal submition of the manuscript. We also checked extensively the manuscript and some expression has been corrected and highlighted with the blue color. We changed the Figure 10, as you suggested, because of the low resolution.
This manuscript is a resubmission of an earlier submission. The following is a list of the peer review reports and author responses from that submission.
Round 1
Reviewer 1 Report
The manuscript describes a software application built based on the Kinect sensor, which is designed for monitoring food intake actions. The reviewer's opinion is that the paper does not show enough novelty for acceptance.
The quality of the text must be improved, there are many cases were wrong connection words are used, e.g., lines 6, 99, 141, 311, 313, etc. Moreover, the text is hard to read, and it is brimming with a wide range of details.
There are many details missing from the presentation.
1: The literature review is not well done; many state-of-the-art papers are missing. Which version of the Kinect sensor is used in this application? Present a set of arguments regarding the selection. What is the frame resolution? Since RGB+D information is captured by the Kinect sensor, why is the proposed application processing only the depth information and not the RGB+D information? If RGB is not used, why not use other time-of-flight sensors with a better quality? The depth map information provided by depth sensors is heavily affected by noise. How is the proposed applications denoising the depth information? Please provide more algorithmic details regarding the propose applications. The steps of the proposed methods are not well designed. Line 207. How is it possible that the previous version had a several hours runtime. How was the complexity of the proposed application reduced? There is no comparison with a state-of-the-art method. A discussion and analysis regarding the results is missing. What is the accuracy of the proposed application? What is the “*” sign stands for? Why in most of the cases the end frame is the last frame? How come the experiments were designed in such a way? Please pay more attention to the list of references. Please remove the “neural network” keyword, since the manuscript contains only one sentence (lines 103-104) which mentions that unsupervised learning is employed to train a neural network, and no other details are presented. I recommend to rewrite the abstract. Please avoid using the expression “in this document”.
Author Response
Thanks for your comments. We have checked all the English expressions and extensively corrected and revised the manuscript, where the new parts are highlighted in blue colour.
We extended the literature to the related works with respect to the present paper.
We introduced the version of the Kinect sensor (v1), because this version was available for the tests in the previous work, so we continued with this version to optimize what has already be done.
The frame resolution is introduced (1920x1080 px).
We have specified that the RGB video acquired is used as a ground-truth (introduced in the Table 1, Table 3 and the Figure 14 highlights the comparison between the ground-truth and the actions detected).
To de-noise the depth information we used a median filter. It was mentioned in the paper but we remarked this in other parts of the revised work (line 138, 328).
To better clarify the steps of the method we introduced the algorithm block diagram in Figure 5.
The previous version had a several hours runtime because missed a software for Kinect for data storage only when the subjects were in the scene and also because it had never been tested with a large number of frames.
The system performance are now introduced with calculation of MAE and MRE, the “*” sign, which indicated the error with respect the ground-truth, is removed.
In most of the cases the end frame is the last frame because the end frame is the last frame processed but not the last saved. For the tests with 500 frames, they matches because the sequence gets interrupted by setting the number of frame to be processed in the developed software interface for Kinect. The software interface is developed with the possibility to choice if to set the acquisition time.
The abstract has been rewritten and the SOM Ex algorithm, the neural network used to data processing, has been better explained.
Reviewer 2 Report
This is an interesting topic. The result could be a useful application. However, there are several parts needed to be improved:
There were many existing algorithms/methods which were developed to identify human gestures/poses. However, they were not mentioned in this paper. Structure of the paper should have shown which parts for literature reviews/background study and which parts of the proposed methods. Despite that the paper had a long text to describe the methods implemented, the reader is still unclear why they should be done like those. There is no performance evaluation or comparing the result with existing methods.Author Response
Thanks for your comments. We have checked all the English expressions and extensively corrected and revised the manuscript, where the new parts are highlighted in blue color.
We extended the literature to the related works with respect to the present paper.
We have specified that the RGB video acquired is used as a ground-truth (introduced in the Table 1, Table 3 and the Figure 14 highlights the comparison between the ground-truth and the actions detected), to better understand the developed approach.
To better clarify the steps of the method we introduced the algorithm block diagram in Figure 5.
The system performance are now introduced with calculation of MAE and MRE, the “*” sign, which indicated the error with respect the ground-truth, is removed.
Reviewer 3 Report
1. The main technical contribution should be better clarified not only in respect to previous publications from the authors but also in comparison with the state-of-art. The authors should reduce the descriptions of previous work and provide a more extensive and detailed descriptions of the technical details of the proposed method.
2. The reviewer consider that the proposed method is rather simplified and there are several parameters that shall be determined. Based on the descriptions, it is rather difficult to understand how the different parameters have been determined. It is also difficult to understand how the selected parameters may be adjusted depending on the different conditions in the room (e.g. light conditions, etc.)
3. The reviewer suggest to increase the number of subjects as well to vary the room conditions in order to better verify the effectiveness of the proposed method. As for the comparison between the Table 1 and 2, it is rather difficult to understand the meaning of "..achieves almost the same results..". The authors may consider analyze the experimental results by means of applying a statistical analysis.
4. It is not clear if the results shown in Table 1 and 2 only represent a single subject or if it is the mean average from the seven subjects,
5. The abstract and conclusions should be revised accordingly after revising the above items.
Author Response
Thanks for your comments. We have checked all the English expressions and extensively corrected and revised the manuscript in accord to your statements. The new parts are highlighted in blue color.
We extended the literature to the related works with respect to the present paper.
We have specified that the RGB video acquired is used as a ground-truth (introduced in the Table 1, Table 3 and the Figure 14 highlights the comparison between the ground-truth and the actions detected).
To better clarify the steps of the method we introduced the algorithm block diagram in Figure 5.
The system performance are now introduced applying a statistical analysis as you suggest, with the calculation of MAE and MRE, the “*” sign, which indicated the error with respect the ground-truth, is removed.
In most of the cases the end frame is the last frame because the end frame is the last frame processed but not the last saved. For the tests with 500 frames, they matches because the sequence gets interrupted by setting the number of frame to be processed in the developed software interface for Kinect. The software interface is developed with the possibility to choice if to set the acquisition time.
The Abstract and the Conclusions has been rewritten.
Round 2
Reviewer 1 Report
The text is hard to read, e.g., “[…]The novelty introduced in this paper is a data storage system through a developed Kinect software interface and an automatic identification of initial and final frame for food intake actions detection.[…]”
The text still contains a lot of typos.
Please introduce the mathematical notation for: Mean Absolute Error (MAE) and Mean Relative Error (MRE). What is the difference between them? MRE is computed relative to which metric/value?
The text in Section 5 should refer more to Figure 5. Moreover, please redraw the figure. The arrows are not visible. Use mathematical notations for the inequalities checked in several blocks. Use the same font or present an explanation why in some blocks it is necessary to use a smaller font.
Please avoid inserting in a manuscript figures generated using a print-screen tool. In MATLAB, you can use the “print” function to save the figures to .eps files which can be then inserted in the LaTeX source file. Moreover, please replace Figures 11 and 12 with corresponding tables.
Figure 14: The font size of the title on the left side is too small. Why did you use red color for the title of the figure on the right side?
Section 2: "The fitting algorithm" - please change the title of this section.
Reviewer 2 Report
The updated version's layout was improved. However, the work itself should be considered:
No clue how to obtain ground truth for Start Frame and End Frame. How many cases, e.g. participants, scenarios, in the dataset were used in this work? The work was introduced to improve the SOM Ex's prediction. There was, however, very little explanation to claim this. Another minor thing is that several thresholds, e.g. for sitting, closing to the table, were manually set which are not ideal. These might only work for specific dataset.Reviewer 3 Report
The authors should provide include a better critical analysis of the current state-of-art. The improvements proposed by the authors are rather based on well-known methods which are rather quite simplified by defining the respective thresholds under controlled experimental conditions. Therefore, it is rather difficult to evaluate the real effectiveness of the proposed methods while taking into consideration the stochastic characteristics of human motions. The lack of comparison of the presented results with other related work makes even more difficult to understand their main technical contribution.